# No Clinical Symptom Experienced after Consumption of Berry Fruits with Positive RT-qPCR Signals of Human Norovirus

**DOI:** 10.3390/pathogens10070846

**Published:** 2021-07-05

**Authors:** Mohamad Eshaghi Gorji, Malcolm Turk Hsern Tan, Mitchie Y. Zhao, Dan Li

**Affiliations:** Department of Food Science & Technology, Faculty of Science, National University of Singapore, Singapore 117542, Singapore; Mohamad.e.gorji@u.nus.edu (M.E.G.); malcolmtth@u.nus.edu (M.T.H.T.); y.zhao@u.nus.edu (M.Y.Z.)

**Keywords:** norovirus, berry, infectivity, public health

## Abstract

Human noroviruses (hNoVs) are the most important foodborne viruses, and soft berries are one of the most common food sources of hNoV outbreaks and contamination. This paper presents a human volunteer study in order to investigate the correlation between molecular detection results of hNoV in berries with the public health risks. The participants with diverse histo-blood group antigens (HBGAs) phenotypes were required to consume self-purchased berries and meanwhile submit aliquots of the products for reverse transcription-quantitative polymerase chain reaction (RT-qPCR) detection. As a result, none of the 20 participants reported any hNoV infection-like symptoms after six independent consumptions (120 consumptions in total). In contrast, within the 68 berry samples with >1% virus recoveries, 28 samples were detected to be positive for hNoV GI and/or GII (the positive rate at 41%). All of the positive signals were below the limit of quantification (<120 genome copies/g) except one fresh strawberry sample at 252 genome copies/g. It is expected that this study would contribute to the definition of quantitative standards for risk assessment purposes in the future.

## 1. Introduction

Human noroviruses (hNoVs) cause between 73% to greater than 95% of global epidemic non-bacterial gastroenteritis outbreaks and approximately half of all gastroenteritis outbreaks [1]. Historically, fresh produce, especially soft berries, are one of the most common food sources of hNoV outbreaks and contamination events, as reviewed by Bozkurt et al. [2]. For instance, in 2012, a huge outbreak affecting approximately 11,000 people occurred in Germany due to hNoV contaminated strawberries [3]. In 2019, multiple hNoV outbreaks on cruise ships were reported associated with frozen fruits and berries in the United States [4].

Despite the recent breakthroughs [5,6,7], the cultivation of hNoVs in the laboratory remains costly and labor intensive. Moreover, none of the current methods is feasible to measure the virus viability from food samples, mainly due to the limitation of sensitivity. So far, the detection of hNoVs from food samples still relies on molecular techniques, mostly reverse transcription-quantitative polymerase chain reaction (RT-qPCR). RT-qPCR, which is recognized as the gold standard for hNoV detection and proposed in the ISO/TS 15216 methods, cannot differentiate between infectious and non-infectious viruses. Therefore, still there is a need for correlation of presence and levels of hNoV as detected by RT-qPCR in foods such as berries to the actual public health risks.

In this study, in order to evaluate the hNoV contamination in berry products at the retail markets and the actual public health risks, a human volunteer study was conducted, and the participants were required to consume self-purchased berries and meanwhile to submit aliquots of the products for RT-qPCR detection.

## 2. Results and Discussion

Due to the technical limitations, clinical trials are currently the only possibility validating the direct public health risks of the food samples detected with hNoV positive signals by RT-qPCR. As clinical trials are known to be costly and risky [8], instead, we conducted a consumption follow-up study that no berry sample consumed by the participants was intentionally spiked with viruses. The volunteers were requested to consume self-purchased berry products from the retail markets so that the health risks they were exposed to by participating in this study are comparable with their own daily diet. Twenty healthy adults were therefore recruited for the human volunteer study, and each of them consumed 6 berry samples (including fresh and frozen strawberries and raspberries, >50 g berries for each consumption) self-purchased from Singapore retail markets in 6 different weeks during a 3-month period (Jan. to Mar. of 2020). Aliquots (>25 g each) of the berry products were submitted by the participants to our laboratory after each consumption for RT-qPCR detection.

The rationale of designing this study was based on the large scales hNoVs screening from berry products conducted and reported in recent years. For instance, in the United Kingdom retail markets, 7/310 (2.3%) fresh raspberries samples and 10/274 (3.6%) samples of frozen raspberries were tested to be hNoV-positive [9]. In China (Heilongjiang province), among 900 frozen and 900 fresh domestic retail berry samples, the prevalence of hNoV was reported to be 9% (81/900) and 12.11% (109/900) [10]. Therefore, a total of 120 berry samples were collected, expecting a likelihood of hNoV occurrence in the berries. As a result, only 68 out of the 120 samples were with virus recoveries >1% (criteria as defined by ISO 15216-2 for a valid detection, calculated based on the detection of spiked process control virus MS2), possibly because the berry samples have been repeatedly frozen and de-frozen from the purchase, consumption, storage, and transportation to the laboratory by individual participants and storage in the laboratory before detection. The berry components such as polyphenols and pectin could have been released to large extents affecting the recovery of viruses as suggested by Zhao and Li [11]. Therefore, future studies improving the detection efficiencies of enteric viruses from berry fruits are still in need. In this study, within the 68 samples with virus recoveries >1%, 28 samples were detected with hNoV presence (41%, 19 samples with hNoV GI only, 2 samples with hNoV GII only, 7 samples with both hNoV GI and GII, Table 1), exceeding most of the reported prevalence of hNoV in berry fruits as shown above. The positive samples were originated from multiple countries, including Serbia, the USA, the UK, Chile, Korea, Japan, New Zealand, Mexico, and South Africa. Due to the limitation of sample numbers, it was not possible to attribute the contamination with different sample types and origins.

In contrast, none of the participants reported any hNoV infection-like symptoms such as diarrheal and/or vomiting after consuming the berries with positive RT-qPCR signals of hNoVs. The susceptibility of an individual towards hNoV infection is known to be closely associated with his/her HBGA phenotype [12,13], and therefore, the HBGA phenotypes were measured for all of the participants in this study. Specifically, within the 16 participants who consumed berries with positive hNoV RT-qPCR signals, there were four type A, three type B, two type AB, and two type O individuals, which were all confirmed to be secretors by the detection of Fucα1-2Gal-R (Table 1). Additionally, there were two secretors being positive for Fucα1-2Gal-R, but their ABO(H) types were not identified (Table 1) as the antibodies used in this study were not able to cover all of the types (anti-B [BG 3] is specific for type 2 chain, anti-H type 1 [BG 4] is specific for type 1 chain). Three individuals were considered non-secretors since there was no reaction of their saliva samples towards UEA-I or ABO(H) antibodies (Table 1). For the Lewis antigens, Le^y+^ was the most common, accounting for about 94% (15/16), followed by Le^b+^ (88%, 14/16), Le^a+^ (56%, 9/16), and Le^x+^ (50%, 8/16) (Table 1). This distribution agrees with previous reports over the HBGA phenotypes of the Chinese ethnicity group [14]. In short, the HBGA phenotypes of the participants were diverse, and it was not possible to attribute the absence of clinical symptoms with the HBGA profiles of the participants.

Prior asymptomatic hNoV infection before the berry consumption, which resulted in acquired immunity of the participants, might be another explanation of the participants being resistant to hNoV exposure. Kobayashi et al. [15] recently reported a norovirus detection rate of 2.5% in asymptomatic adults by investigating15,532 participants. Indeed, the presence of low levels of hNoV genomes (GI or GII, Ct values ≥ 35) in the stool samples was observed in 10 out of the 16 participants before they consumed berries with positive hNoV RT-qPCR signals (Table 1). Interestingly, the asymptomatic carrying of hNoV GII genomes was also observed in one of the three non-secretors, which is possible but rare based on previous reports [16]. However, several lines of evidence indicate that hNoVs can antagonize or evade host immune responses, including the absence of long-lasting immunity elicited during a primary hNoV exposure [17]. Thus, it remains unclear whether the low levels of virus genome presence in stool samples could indicate protective immunity acquired by the participants.

Attempts to correlate the human health risk of NoV in foods by comparing the hNoV RNA levels detected by RT-qPCR and the illness reports from people after the food consumption is scarce. Lowther et al. [18] collected both “non-outbreak-related samples” from commercial shellfish production areas within the British Isles and tested in the laboratory and “outbreak-related samples” which were unambiguously linked to one or more cases of hNoV or hNoV-type illness in consumers were obtained from the UK National Reference Laboratory. A statistically significant difference between hNoV levels in the two sets of samples was observed, and the geometric mean of the levels in outbreak samples (1048 copies per g) was almost one order of magnitude higher than for positive non-outbreak-related samples (121 copies per g). Very interestingly, in this study, all of the positive signals from the 26 positive berry samples were below the limit of quantification (<120 genomic copies/g) except for one fresh strawberry sample at 252 genomic copies/25g. In other words, all of the positive signals identified in this study were with the same magnitude of the non-outbreak-related positive samples as reported by Lowther et al. [18]. Therefore, we assume the main reason for the absence of clinical symptoms after the consumption of berry fruits with positive RT-qPCR signals of hNoVs is the low quantities of viruses in the berry fruits, as the naturally present virus population might have a large inactive portion due to various environmental stress in the food chain [19,20].

## 3. Materials and Methods

### 3.1. Human Volunteer Study Organization

The human volunteer study was approved by Institutional Review Board at the National University of Singapore (NUS-IRB REFERENCE CODE: H-19-047). Twenty healthy adults between 21 and 40 years old, who do not suffer from severe chronic diseases or health problems, digestive issues (vomiting, diarrheal, etc.) at the time of participation, or are allergic to berry or berry products, were recruited. The ethnicity group of all the 20 participants were Chinese. Each participant supplied a saliva sample (about 1 mL) and a stool sample (about 1 g) in order to determine the histo-blood group antigen (HBGA) phenotype and to determine whether this participant is an asymptomatic NoV carrier before this study. During a 3-month period (from January to March 2020), each subject was required to purchase 6 berry samples (strawberries or raspberries, fresh or frozen berries) of different origins/brands from Singapore retail markets in six different weeks. Each time after purchase, the subject should consume >50 g of the berries within 24 h. The berries can be consumed in different ways (directly consumed or with milk, cereal, etc.) but cannot be heated/further cooked. For each sample after consumption, the subject should bring back to us >25 g berries from the same package (to be kept in refrigerator or freezer, or on ice if the sample cannot reach the lab within one hour at room temperature) together with the product information (date of purchase, local retailer, brands and origins). On every occasion of consumption of berry products, the participants should report any disease symptoms (diarrhea and/or vomiting) if occurring due to consumption of the berries within 48h.

### 3.2. Detection of Fucα1-2Gal and HBGA Phenotyping in Saliva Samples

The saliva samples collected from the participants were boiled at 95 °C for 10 min, followed by centrifugation at 10,000× *g* for 5 min. The supernatant was collected and diluted at 1:300 by carbonate: bicarbonate buffer (pH 9.4) and was used to coat 96-well microtiter plates (100 μL/well) at 4 °C overnight. After being washed once with phosphate-buffered saline (PBS) containing 0.05% Tween 20 (PBS-T), the plate was blocked with 5% nonfat dried milk (Blotto, 200 μL/well) at 37 °C for 1 h. In order to detect Fucα1-2Gal-R, which is specifically present in secretor but not in non-secretor saliva, HRP-conjugated Ulex europaeus agglutinin (UEA-I, Sigma-Aldrich, St Louis, MO, USA) diluted at 1:3200 (starting conc. 1 mg/mL) was added and incubated for 1 h at 37 °C. After being washed three times with PBS-T, the presence of Fucα1-2Gal-R was detected with a TMB (3,3′,5,5′-tetramethylbenzidine) kit (Sigma-Aldrich), and the signal intensities (the optical density at 450 nm (OD450)) were read with a Multiskan Sky plate reader (Thermo Fisher Scientific, Shanghai, China). In order to determine the HBGA phenotype of the saliva samples, the blocked wells were added with 1:200 diluted HBGA monoclonal antibody anti-A (BG 2), anti-B (BG 3), anti-H type 1 (BG 4), anti-Lewis a (BG 5), anti-Lewis b (BG 6), anti-Lewis x (BG 7), and anti-Lewis y (BG 8) (MAb; Covance, Emeryville, CA, USA) and then incubated for 1 h at 37 °C. After being washed three times with PBS-T, the HBGA phenotype was detected with horseradish peroxidase conjugated with goat anti-mouse IgG (1:1500; Sigma-Aldrich, Chesterfield, MO, USA) for type A, H, and Lewis a, and goat anti-mouse IgM (1:1500; Sigma-Aldrich USA) for type B and Lewis b, x and y. Horseradish peroxidase activity was detected with a TMB (3,3′,5,5′-tetramethylbenzidine) kit (Sigma-Aldrich), and the signal intensities (the optical density at 450 nm (OD450)) were read with the Multiskan Sky plate reader. The cutoff value was taken as two times the absorbance value of the negative controls as previously performed by Nordgren et al. [21].

### 3.3. Virus Extraction Procedures from Berry Samples Based on ISO 15216-2

The virus extraction from berry samples was performed based on ISO 15216-2, as described previously [11]. Twenty-five g of berry sample in a stomacher bag with filter compartment (3 M Corporation, USA) was spiked with 10 μL of MS2 suspension (10^7^ PFU in 10 μL MS2 spike) and incubated at 25 °C for 30 min, shaking at 60 oscillationˑmin^−1^ with 40 mL of Tris Glycine Beef Extract buffer (TGBE) and 30 U of pectinase (Sigma-Aldrich, USA) from *Aspergillus niger*. The pH of the buffer in the stomacher bag was measured and adjusted to 9.0–10.0 every 10 min. The filtered solution was clarified by centrifugation at 10,000× *g* for 30 min at 4 °C. The pH of the supernatant was adjusted to 7.0, and polyethylene glycol (PEG) 8000/NaCl solution (Sigma-Aldrich, USA) was added for precipitation for 60 min at 4 °C, shaking at 60 oscillationˑmin^−1^. After centrifugation at 10,000× *g* for 30 min at 4 °C, the pellet was collected and resuspended in 1 mL PBS. An equal volume of chloroform/butanol (1:1) was added to the solution, vortexed vigorously and centrifuged at 10,000× *g* for 15 min at 4 °C. The aqueous phase (approximately 1 mL) was collected as the virus extract.

### 3.4. Virus Extraction from Stool Samples

The stool samples were resuspended in PBS at a ratio of 1:9, vortex for at least 1 min, and clarified by centrifugation at 10,000× *g* for 30 min at 4 °C. The supernatants were collected to proceed with RNA extraction and RT-qPCR analyses for hNoV GI and GII.

### 3.5. RNA Extraction and RT-qPCR Analyses

For the RNA extraction, the RNeasy Mini Kit (Qiagen, Germany) was used in the study following the manufacturer’s instructions (30 μL RNA extracted from 100 μL of the virus extract). The RT-qPCR analyses were performed by adding 5 μL of extracted viral RNA in a 15 μL reaction mix using GoTaq^®^ Probe 1-Step RT-qPCR System (Promega, Madison, WI, USA). Table 2 summarizes the primers and probes used in this study with respect to the final concentrations used. The following general cycling conditions were applied: reverse transcription at 45 °C for 15 min, pre-denaturation at 95 °C for 10 min, followed by 45 cycles of denaturation at 95 °C for 15 s and annealing/extension with conditions varied from different virus targets (Table 2).

Double-stranded DNA (dsDNA) containing the specific primer-probe binding sites (Table 2) were synthesized for NoVs (GI and GII) and cloned into the pGEM-T Vector (Promega), resulting in the NoV-GI and NoV-GII plasmids as positive controls. The plasmid concentration was determined by photospectroscopy at 260 nm using the BioDrop DuoTM spectrophotometer (BioDrop, United Kingdom). The limit of detection (LOD) was defined as 5 genome copies/RT-qPCR reaction, which is the lowest level being detected robustly in our preliminary experiments (data not shown). Ten-fold serial dilutions of positive control plasmids were used to prepare the standard curves. The limit of quantification (LOQ) was regarded as 50 genome copies/RT-qPCR reaction since good linearity of the standard curves were only obtained within the range between 50 to 5 × 10^6^ genome copies/RT-qPCR reaction (Figure 1, R^2^ > 0.999 for both hNoV GI and GII). Negative process controls were included in each batch of virus extraction and the RNA extraction and RT-qPCR afterwards. Negative RT-qPCR controls (water as RT-qPCR template) were included in each RT-qPCR run.

A standard curve of MS2 RT-qPCR detection was prepared between the RT-qPCR cycle threshold (Ct) values and MS2 RNA ten-fold serial dilutions in order to calculate the virus recovery rates. Virus recovery was accessed using MS2 with an acceptability criterion of ≥1%.

## 4. Conclusions

In conclusion, the berry consumption follow-up study with human volunteers suggested the common presence of non-infectious hNoVs on the berry products at the retail markets. A conscious interpretation must take into consideration of the actual meaning of molecular detection signals when trying to correlate the screening results with the public health risks. It is expected that this study will contribute to the definition of quantitative standards for risk assessment purposes in the future.

## Figures and Tables

**Figure 1 pathogens-10-00846-f001:**
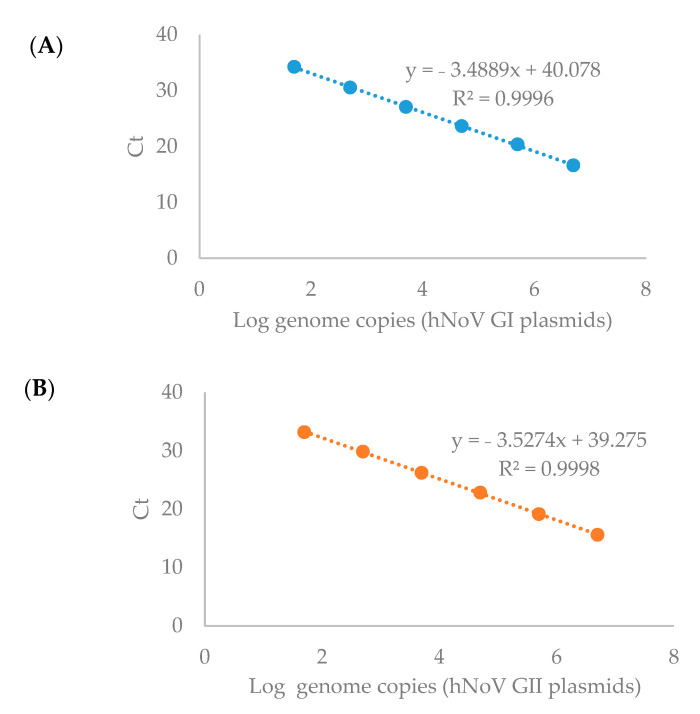
Standard curves for the RT-qPCR detection of hNoV GI (**A**) and GII (**B**).

**Table 1 pathogens-10-00846-t001:** The saliva phenotypes and prior hNoV infection detection of the cohort who consumed berries with positive hNoV RT-qPCR signals.

Saliva Phenotypes of the Cohort Consumed Berries with Positive hNoV RT-qPCR Signals								RT-qPCR Detection (Ct Values) of hNoVs from Stool Samples of the Cohort before Berry Consumption		RT-qPCR Detection Results (Ct Values) of hNoVs (GI and GII) from Berry Samples with Recovery Rates Higher than 1%
Fucα1-2Gal-R	A	B	H1	Le^a^	Le^b^	Le^x^	Le^y^	GI	GII	
+	+	-	-	+	+	-	+	39.7	NA	Fresh raspberry (GI 37.1); Fresh strawberry (GI 37.3)
+	+	-	-	-	+	+	+	NA	36.8	Frozen raspberry (GI 35.3, GII 35.3)
+	+	-	-	+	-	+	+	39.9	NA	Fresh strawberry (GI 36.0)
+	+	-	-	+	+	+	+	38.6	NA	Frozen raspberry (GI 35.3, GII 34.6)
+	-	+	-	+	+	-	+	NA	NA	Frozen strawberry (GI 34.4); Frozen raspberry (GI 38.7)
+	-	+	-	-	+	-	+	NA	36.1	Fresh strawberry (GI 39.7); Fresh strawberry (GII 39.5)
+	-	+	-	-	+	-	+	NA	NA	Fresh strawberry (GI 39.5); Frozen strawberry (GI 39.6)
+	+	+	-	+	+	-	+	38.5	NA	Fresh raspberry (GI 38.5)
+	+	+	-	+	+	+	+	NA	NA	Fresh strawberry (GI 40.0, GII 38.0); Fresh strawberry (GI 40.0)
+	-	-	+	-	-	-	+	NA	34.7	Fresh strawberry (GI 39.1)
+	-	-	+	+	+	+	+	NA	NA	Fresh strawberry (GI 38.6); Frozen strawberry (GI 34.9, GII 37.2)
+	-	-	-	-	+	-	+	NA	38.3	Fresh strawberry (GI 35.5); Fresh strawberry (GI 39.7); Fresh raspberry (GI 39.1)
+	-	-	-	-	+	-	+	NA	37.3	Frozen strawberry (GI 39.5)
-	-	-	-	+	+	+	+	NA	37.7	Frozen raspberry (GI 34.1); Fresh strawberry (GI 38.5, GII 36.8); Fresh strawberry (GI 32.9, GII 36.2); Fresh strawberry (GI 38.9)
-	-	-	-	+	+	+	+	NA	NA	Fresh strawberry (GI 34.8)
-	-	-	-	-	+	+	-	NA	NA	Fresh strawberry (GI 37.7, GII 36.0); Fresh strawberry (GI 38.2)

**Table 2 pathogens-10-00846-t002:** RT-qPCR primers and probes, PCR annealing and extension conditions, and plasmid insert sequences used in this study.

Target	Sequence of Primer, Probe and Plasmids Inserts (5′-3′)		PCR Annealing and Extension Conditions	References
hNoV GI	Forward Primer	CGC TGG ATG CGN TTC CAT	Annealing: 60 °C, 30 s Extension: 72 °C, 30 s	ISO15216:2017
	Reverse Primer	CCT TAG ACG CCA TCA TCA TTT AC		
	Probe	FAM-TGG ACA GGA GAY CGC RAT CT-TAMRA		
	Plasmid Insert	CGCTGGATGCGCTTCCATGACCTCGGATTGTGGACAGGAGATCGCGATCTTCTGCGGATCCGAATTCGTAAATGATGATGGCGTCTAAGG		
hNoV GII	Forward Primer	ATG TTC AGR TGG ATG AGR TTC TCW GA	Annealing: 6 °C, 30 s Extension: 72 °C, 30 s	ISO15216:2017
	Reverse Primer	TCG ACG CCA TCT TCA TTC ACA		
	Probe	FAM-AGC ACG TGG GAG GGC GAT CG-TAMRA		
	Plasmid Insert	ATGTTCAGATGGATGAGATTCTCAGATCTGAGCACGTGGGAGGGCGATCGCAATCTGGCTCGGATCCCCAGCTTTGTGAATGAAGATGGCGTCGA		
MS2	Forward Primer	GCT CTG AGA GCG GCT CTA TTG	Annealing: 60 °C,1 min	[11]
	Reverse Primer	CGT TAT AGC GGA CCG CGT		
	Probe	FAM-CCG AGA CCA ATG TGC GCC GTG-TAMRA		

## Data Availability

The data presented in this study are available on request from the corresponding author.

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
