# Peer review of "No Clinical Symptom Experienced after Consumption of Berry Fruits with Positive RT-qPCR Signals of Human Norovirus"

_pathogens, 2021, doi:10.3390/pathogens10070846_

Round 1

Reviewer 1 Report

Gorji et al. describe a human volunteer study to investigate a prevalence risk of hNoV contaminated berries in retail markets. In the study, volunteers, whose HBGA status and asymptomatic hNoV infection were determined, consumed self-purchased berries and sent its aliquots to the laboratory to detect hNoV by RT-qPCR. The authors show that no clinical symptom was reported, though the RT-qPCR detected hNoV from 28 out of 120 samples. Finally, the authors reason that the presence of hNoV below the limit of quantification (<12- genomic copies) in the berries could lead to the silent. This reviewer agrees that the importance of the correlation between hNoV contaminated foods and symptom caused by the foods and think comments pointed below would improve this manuscript before publication.

Major Comment

The authors show the standard curves for RT-qPCR detection of GI and GII hNoVs in Figure 1 and explain that the LOQ in this study was regarded as 50 genome copies/RT-qPCR reaction. On the other hand, RT-qPCR detection results in Table 1 are represented by Ct values, some of which seems to be below the LOQ range. This reviewer think that the RT-qPCR detection results in Table 1 should be represented by copy number and the numbers below LOQ could be described as like undetected.

Minor Comments

Line 32: The refence 5, which is a review, could be replaced with the original paper (Ettayebi et al, Science, 2016)

Line 66: Please provide a short explanation here for how the virus recovery was determine, though it is in Materials and Methods. Readers might misunderstand the >1% is a number of hNoV recovery rate.

Line 72: How did the authors get the high prevalence rate of hNoV in berry in this study, the 41%? This reviewer think the total 120 samples should be used to calculate the hNoV prevalence rate; 28/120 samples = 23%.

Lines 71-73: Please check the numbers of samples in this sentence. The 26 samples with GI, 9 samples with GII, and 7 samples with GI/GII might be totally 42 samples.

Line 113: Please remove "log" between "252" and "genomic copies".

Line 137: Please replace "fridge" with "refrigerator".

Author Response

Reviewer 1:

Gorji et al. describe a human volunteer study to investigate a prevalence risk of hNoV contaminated berries in retail markets. In the study, volunteers, whose HBGA status and asymptomatic hNoV infection were determined, consumed self-purchased berries and sent its aliquots to the laboratory to detect hNoV by RT-qPCR. The authors show that no clinical symptom was reported, though the RT-qPCR detected hNoV from 28 out of 120 samples. Finally, the authors reason that the presence of hNoV below the limit of quantification (<12- genomic copies) in the berries could lead to the silent. This reviewer agrees that the importance of the correlation between hNoV contaminated foods and symptom caused by the foods and think comments pointed below would improve this manuscript before publication.

Major Comment

The authors show the standard curves for RT-qPCR detection of GI and GII hNoVs in Figure 1 and explain that the LOQ in this study was regarded as 50 genome copies/RT-qPCR reaction. On the other hand, RT-qPCR detection results in Table 1 are represented by Ct values, some of which seems to be below the LOQ range. This reviewer think that the RT-qPCR detection results in Table 1 should be represented by copy number and the numbers below LOQ could be described as like undetected.

Author's Response: Thanks for the comments. Unfortunately, as explained in the text: “in this study, all of the positive signals from the 26 positive berry samples were below the limit of quantification (<120 genomic copies/g) except for one fresh strawberry sample at 252 genomic copies/25g”. Thus it was not possible to present copy numbers in Table 1. Extrapolation of the standard curves to levels below LOQ is not precise and thus should not be done based on our research experience.

Minor Comments

Line 32: The refence 5, which is a review, could be replaced with the original paper (Ettayebi et al, Science, 2016).

Author's Response: Done.

Line 66: Please provide a short explanation here for how the virus recovery was determine, though it is in Materials and Methods. Readers might misunderstand the >1% is a number of hNoV recovery rate.

Author's Response: Done.

Line 72: How did the authors get the high prevalence rate of hNoV in berry in this study, the 41%? This reviewer think the total 120 samples should be used to calculate the hNoV prevalence rate; 28/120 samples = 23%.

Author's Response: A virus recovery of >1% is a criteria as defined by ISO 15216‑2 for a valid detection (added in Line 65 as clarification). Therefore, the prevalence of virus can only be calculated based on the 68 valid samples.

Lines 71-73: Please check the numbers of samples in this sentence. The 26 samples with GI, 9 samples with GII, and 7 samples with GI/GII might be totally 42 samples.

Author's Response: Sorry for the misunderstanding. The 7 samples with both GI and GII were included in the 26 GI and 9 GII samples. We have modified to be clarified now.

Line 113: Please remove "log" between "252" and "genomic copies."

Author's Response: Done. Sorry for the typo.

Line 137: Please replace "fridge" with "refrigerator."

Author's Response: Done. 

Reviewer 2 Report

The paper by Gorji et al reports a human volunteer study where healthy adults consumed self-purchased berries, naturally-contaminated or not by Human Norovirus, with the aim to investigate the correlation between viral load in the berries and gastroenteritis symptoms in volunteers.

The study design is interesting, as it is very representative of the real public health risk associated to berry consumption. The data suggest that NoV-contaminated berries with very low viral load are not associated with clinical symptoms in healthy adults, which is in line with previous work on contaminated shellfish. This type of study is rare and very important for the field.

However some results are quite surprising and could be discussed more :

  • The very high prevalence of HuNoV-positive berries (41%, here in Singapore, vs 9-12% in China):
    • How could this discrepancy be explained?
    • Is there a reason to suspect that berries sold in Singapore are more at risk of contamination?
    • What was their origin?
  • The high proportion of NoV-positive stool samples from volunteers at the beginning of the study: 4 volunteers were positive for GI (20%), 6 for GII (30%) which represents a very high prevalence for asymptomatic NoV shedding. For comparison, a recent study by Kobayashi et al (Clin Microbiol Infect 2021) reports a prevalence of 2,5% (NoV GI + GII) among healthy adults in Japan (and could be cited). Could the authors discuss this point? Would it be possible to assess the anti-NoV immunity it serum samples from (at least some of) the study volunteers, to confirm NoV exposure? Were fecal samples collected at a later time point in the study to check for viral shedding, and not only symptomatic infection?
  • The detection of NoV GII in a fecal sample from a secretor-negative volunteer is unusual. This point should be discussed also.

As expected for naturally contaminated berries and stools from asymptomatic individuals, NoV loads are very low when detected. Thus, to completely exclude that the high NoV prevalence be due to some false positive RTqPCR results, some more details are needed in the material and methods section. The limit of quantification is set at 50 gc/reaction, but what is the limit of detection of the technique? Has it been assessed by the authors (in berries and/or in stools)? If not what is the cut-off Ct value, and how was it selected ? What is the nature of the negative process control used for each extraction? How many replicates were done for each sample (RT-qPCR replicates or extraction replicates…)? Was a go-forward process applied when analyzing the stool and berries samples, or other means to prevent contaminations?

A few spelling corrections :

  • L38: ISO 15.216 (not 15,216)
  • L80: participants *who* consumed berries

Author Response

Reviewer 2:

The paper by Gorji et al reports a human volunteer study where healthy adults consumed self-purchased berries, naturally-contaminated or not by Human Norovirus, with the aim to investigate the correlation between viral load in the berries and gastroenteritis symptoms in volunteers.

The study design is interesting, as it is very representative of the real public health risk associated to berry consumption. The data suggest that NoV-contaminated berries with very low viral load are not associated with clinical symptoms in healthy adults, which is in line with previous work on contaminated shellfish. This type of study is rare and very important for the field.

However some results are quite surprising and could be discussed more :

  • The very high prevalence of HuNoV-positive berries (41%, here in Singapore, vs 9-12% in China):
    • How could this discrepancy be explained?
    • Is there a reason to suspect that berries sold in Singapore are more at risk of contamination?
    • What was their origin?

Author's Response: Extra information has been added to the text: “The positive samples were originated from multiple countries including Serbia, USA, UK, Chile, Korea, Japan, New Zealand, Mexico, and South Africa. Due to the limitation of sample numbers, it was not possible to attribute the contamination with different sample types and origins.” Based on my many years performing prevalence study of virus from foods, this is not surprising as 1) the sample number is very small thus not representative; 2) the current (standardized) detection method still suffers from many uncertainties and the performance from different laboratories vary a lot; 3) as indicated by this study, a positive molecular detection result, especially in low levels, may not represent public health risks.

  • The high proportion of NoV-positive stool samples from volunteers at the beginning of the study: 4 volunteers were positive for GI (20%), 6 for GII (30%) which represents a very high prevalence for asymptomatic NoV shedding. For comparison, a recent study by Kobayashi et al (Clin Microbiol Infect 2021) reports a prevalence of 2.5% (NoV GI + GII) among healthy adults in Japan (and could be cited). Could the authors discuss this point? Would it be possible to assess the anti-NoV immunity it serum samples from (at least some of) the study volunteers, to confirm NoV exposure? Were fecal samples collected at a later time point in the study to check for viral shedding, and not only symptomatic infection?

Author's Response: Thanks for the comments and we completely agree with the reviewer. Unfortunately it is not possible to collect serum or stool samples from the participants for this study anymore as it was performed ~2 years ago and the data will not be valid. Our IRB is expired as well. We would definitely take it into consideration for our future research. As for the comments over asymptomatic NoV prevalence, we have added the study as suggested to support our data. However, again we believe this small sample number limits the representativeness to compare with other studies.

  • The detection of NoV GII in a fecal sample from a secretor-negative volunteer is unusual. This point should be discussed also.

Author's Response: As suggested it is now added in the discussion: “Interestingly, the asymptomatic carrying of hNoV GII genomes was also observed in one of the three non-secretors, which is possible but rare based on previous reports [16]”.

  • As expected for naturally contaminated berries and stools from asymptomatic individuals, NoV loads are very low when detected. Thus, to completely exclude that the high NoV prevalence be due to some false positive RTqPCR results, some more details are needed in the material and methods section. The limit of quantification is set at 50 gc/reaction, but what is the limit of detection of the technique ? Has it been assessed by the authors (in berries and/or in stools)? If not what is the cut-off Ct value, and how was it selected? What is the nature of the negative process control used for each extraction ? How many replicates were done for each sample (RT-qPCR replicates or extraction replicates…)? Was a go-forward process applied when analyzing the stool and berries samples, or other means to prevent contaminations?

Author's Response: We understand the concern of the reviewer. We have now added information over LOD in the text: “The limit of detection (LOD) was defined as 5 genome copies/RT-qPCR reaction, which is the lowest level being detected robustly in our preliminary experiments (data not shown).” As stated in the text, “Negative process controls were included in each batch of virus extraction and the RNA extraction and RT-qPCR afterwards. Negative RT-qPCR controls (water as RT-qPCR template) were included in each RT-qPCR run”. Based on the limitation of berry sample size we received, only one detection was done for each sample. And yes a go-forward process was applied in our laboratory, we have 3 separate zones for molecular detection, for master mix preparation, sample preparation and addition, and PCR amplification. We have also very strict precautions to prevent PCR contamination such as using filter-tips, applying specific experiment tools with regular bleach and UV treatment. However in the end I have to admit that even with all of the efforts, it is not possible to eliminate the chance for contamination completely, this is also one of the reasons that one should be very cautious when trying to reply on molecular detection results without extra confirmation step to make public health related interpretations.

A few spelling corrections:

  • L38: ISO 15.216 (not 15,216)

Author's Response: There should be no mark in between of the ISO number 15216.

  • L80: participants *who* consumed berries

Author's Response: Done.

Reviewer 3 Report

The manuscript by Mohamad Eshaghi Gorji et al., entitled “No clinical symptom experienced after consumption of berry 2 fruits with positive RT-qPCR signals of human norovirus”. This manuscript describes investigation the correlation between molecular detection of hNoV in berries and public health risks.

The authors enrolled 20 volunteer healthy adults age range between 21-40. The volunteer enrolled collected stool specimens for hNoV testing before consumption berries. Each volunteer consume berry, either fresh or frozen, 6 times and share half of them to lab for hNoV RT-qPCR in 24hrs. The saliva phenotype of participants whose consume berries with hNoV positive signal were compare the relation with hNoV genotype detected from berries.

Comments:

Norovirus is a leading cause of acute diarrhea disease of all population in the world. Several factors are currently increasing the challenge by norovirus infections to global healthy, such as the rapid rate of the genetic and antigenic evolution of circulating noroviruses, food or water contamination which complicates the public policy and medical treatment. In this study, expected to define quantitative standard for risk assessment. There are few questions:   

  1. On page 4, line 135, the berries can be consumed in different ways (directly consumed, or with milk…..)
    Because the berries were in different format, how to differentiate the hNoV contamination is from berries or from other ingredient or food processing?   
  2. Volunteers were collected stool specimens for hNoV testing only before consume berries. Those participants were follow-up and report if they have clinical symptom (diarrhea and/or vomiting) within 48hrs. Asymptomatic of NoV infection were estimated about 30%. If this study collected all participant stool specimens after consume and perform hNoV testing with genotypes. Then this data might have more comprehensive interpretation from food (berries) to case.
  3. Though the viral load detected from 26 positive berry sample (except 1 ) were below the limit of quantification detection. This result is different from previous report by Lowther et al. However, the differences might due to the natural characteristic of shellfish which can concentrate viruses via their filter mechanism. In this study, the authors might suggest an improvement analysis protocol for berry sample as or defined quantitative standards.

Author Response

Reviewer 3:

The manuscript by Mohamad Eshaghi Gorji et al., entitled “No clinical symptom experienced after consumption of berry 2 fruits with positive RT-qPCR signals of human norovirus”. This manuscript describes investigation the correlation between molecular detection of hNoV in berries and public health risks.

The authors enrolled 20 volunteer healthy adults age range between 21-40. The volunteer enrolled collected stool specimens for hNoV testing before consumption berries. Each volunteer consume berry, either fresh or frozen, 6 times and share half of them to lab for hNoV RT-qPCR in 24hrs. The saliva phenotype of participants whose consume berries with hNoV positive signal were compare the relation with hNoV genotype detected from berries.

Comments:

Norovirus is a leading cause of acute diarrhea disease of all population in the world. Several factors are currently increasing the challenge by norovirus infections to global healthy, such as the rapid rate of the genetic and antigenic evolution of circulating noroviruses, food or water contamination which complicates the public policy and medical treatment. In this study, expected to define quantitative standard for risk assessment. There are few questions:   

1. In page 4 line 135, the berries can be consumed in different ways (directly consumed, or with milk…..)
Because the berries were in different format, how to differentiate the hNoV contamination is from berries or from other ingredient or food processing?   

Author's Response: This is absolutely true. Moreover, we also anticipated that the participants might get hNoV infection in the duration of our tests from other dietary ingredients or even other sources such as water, environment or other individuals. Thus we have planned to sequence the clinical samples of the participants, if symptoms appear, and to compare the sequence with virus sequence detected from the berries (if positive results could be obtained by RT-qPCR detection). Unfortunately as explained in the paper, no participant showed any symptoms and thus this plan was not performed.

2. Volunteers were collected stool specimens for hNoV testing only before consume berries. Those participants were follow-up and report if they have clinical symptom (diarrhea and/or vomiting) within 48hrs. Asymptomatic of NoV infection were estimated about 30%. If this study collected all participant stool specimens after consume and perform hNoV testing with genotypes. Then this data might have more comprehensive interpretation from food (berries) to case.

Author's Response: We completely agree with this comment, will take into consideration for our future research. Here in this study, in order to define our set-up and results precisely, we report our results as “No clinical symptom” instead of no infection.

3. Though the viral load detected from 26 positive berry sample (except 1 ) were below the limit of quantification detection. This result is different from previous report by Lowther et al. However, the differences might due to the natural characteristic of shellfish which can concentrate viruses via their filter mechanism. In this study, the authors might suggest an improvement analysis protocol for berry sample as or defined quantitative standards.

Author's Response: Agree. We have added the following statement in the results and discussion: “Therefore, future studies improving the detection efficiencies of enteric viruses from berry fruits are still in need” (Line 71).